# Classical cadherins evolutionary constraints in primates is associated with their expression in the central nervous system

**Max Petersen**, **Fredy Reyes-Vigil**, **Marc Campo, Juan L. Brusés** *

Department of Natural Sciences, School of Health and Natural Sciences, Mercy University, Dobbs Ferry, New York, United States of America

☯ These authors contributed equally to this work.

* jbruses@mercy.edu

**Data Availability Statement:** All relevant data are within the manuscript and its Supporting information files.

## Abstract

Classical cadherins (CDH) comprise a family of single-pass transmembrane glycoproteins that contribute to tissue morphogenesis by regulating cell-cell adhesion, cytoskeletal dynamics, and cell signaling. CDH are grouped into type I (CDH 1, 2, 3, 4 and 15) and type II (CDH 5, 6, 7, 8, 9, 10, 11, 12, 18, 20, 22 and 24), based on the folding of the cadherin binding domain involved in *trans*-dimer formation. CDH are exclusively found in metazoans, and the origin and expansion of the gene family coincide with the emergence of multicellularity and vertebrates respectively. This study examined the evolutionary changes of CDH orthologs in primates and the factors that influence selective pressure to investigate the varying constraints exerted among CDH. Pairwise comparisons of the number of amino acid substitutions and of the ratio of non-synonymous substitutions per non-synonymous sites (dN) over synonymous substitutions per synonymous sites (dS), show that CDH2, CDH4, and most type II CDH have been under significantly higher negative selective pressure as compared to CDH1, CDH3, CDH5 and CDH19. Evaluation of gene essentiality as determined by the effect of germline deletion on animal viability, morphogenic phenotype, and reproductive fitness, show no correlation with the with extent of negative selection observed on CDH. Spearman's correlation analysis shows a positive correlation between CDH expression levels (E) in mouse and human tissues and their rate of evolution (R), as observed in most proteins expressed on the cell surface. However, CDH expression in the CNS show a significant E-R negative correlation, indicating that the strong negative selection exerted on CDH2, CDH4, and most type II CDH is associated with their expression in the CNS. CDH participate in a variety of cellular processes in the CNS including neuronal migration and functional assembly of neural circuits, which could profoundly influence animal fitness. Therefore, our findings suggest that the unusually high negative selective pressure exerted on CDH2, CDH4 and most type II CDH is due to their role in CNS formation and function and may have contributed to shape the evolution of the CNS in primates.

**Funding:** This work was supported in part by Mercy University Provost's Office Awards and the L. Squitieri fund to M.P, F.R-V, and J.L.B.

**Competing interests:** The authors have declared that no competing interests exist.

## Introduction

The cadherin family of cell adhesion molecules is comprised of over a hundred genes encoding transmembrane proteins playing important roles in tissue morphogenesis by regulating cell-cell adhesion, cytoskeletal dynamics, cell signaling, and gene expression [1–5]. Cadherins are exclusively found in metazoans and their evolutionary origin coincides with the appearance of multicellularity approximately 600 million years ago (MYr) [6]. The emergence of cadherins contributed to the development of complex tissue architectures through the formation of intercellular junctions, providing mechanical support to cell contacts and serving as scaffolding for the assembly of signaling hubs regulating cell shape and behavior [7,8]. Cadherins are characterized by having an extracellular cadherin domain (EC) comprised of ~110 amino acids folded into seven beta strands forming two beta sheets [9–11]. Cadherins are divided into three main branches based on structural similarities and phylogenetic relationships: classical cadherins (CDH), protocadherins, and cadherin-related proteins [12]. CDH constitute a group of 18 single-pass transmembrane glycoproteins with similar topology, comprised of a cadherin N-terminal pro-domain, an ectodomain (ED) with four homologous ECs (EC1 to EC4) and a less related EC5 with three calcium-binding motifs between ECs, and a highly conserved cytoplasmic domain (CD) containing a membrane-proximal and a C-terminal binding site for p120-catenin and beta-catenin respectively [4,11,13–15]. The cleavage of the N-terminal cadherin pro-domain during transport to the cell membrane and calcium-binding are necessary for high-affinity homo and heterophilic trans-dimerization, while the binding of catenins to the CD regulates cytoskeletal dynamics [16–20]. CDH are divided into type I and type II based on the folding of EC1 [15,18,21,22]. Type I CDH (CDH1, CDH2, CDH3, CDH4, and CDH15) have a single tryptophan residue in position 2 (W2) and a His-Ala-Val (HAV) tripeptide in EC1 [19,23]. In contrast, CDH type II (CDH5, CDH6, CDH7, CDH8, CDH9, CDH10, CDH11, CDH12, CDH18, CDH19, CDH20, CDH22, and CDH24) are characterized by having two tryptophan residues (W2 and W4) in EC1 and lacking an HAV peptide [3,12,15,19,22–25]. Both, CDH type I and type II form homo and heterophilic bindings between members of the same type [19,22,24,26,27]. Based on heterophilic binding affinities, CDH type II are subdivided into three specificity groups: A) CDH6, CDH9 and CDH10; B) CDH7, CDH12, CDH18, CDH20 and CDH22; and C) CDH8, CDH11 and CDH24 [24]. The solitary CDH13 has a unique EC1 folding, does not form heterophilic interactions, and is the only cadherin with no transmembrane domain linked to the cell membrane via a glycosylphosphatidylinositol (GPI) moiety [12,28,29]. The expression of distinct CDH repertoires in specific tissues during precise times of development supports specific cell sorting and tissue boundary formation. Varying combinations of CDH type II expressed by distinct neuronal groups affect cell migration, positioning, and formation of neuronal contacts [1,30,31]. In contrast to type I, disruption of CDH type II expression during embryogenesis results in subtle morphogenic defects due to improper segregation of neuronal pools and defective selection of synaptic partners [31–35].

The enlargement of the cadherin gene family coincides with the emergence of vertebrates, suggesting that cadherins contributed to the organization of vertebrates' body plan and the formation of architecturally complex tissues including the CNS [36]. Due to their importance in establishing neuronal identity and connectivity, CDH may have contributed to the recent evolution of neural structures and the acquisition of the larger repertoires of neuronal connections observed in primates. Varying degrees of conservation between cadherins orthologs have been noticed throughout vertebrate's phylogeny [3,12,37]; however, whether cadherins have been under varying evolutionary constraints during the evolution of primates has not been examined. A better understanding of the evolution of CDH may contribute to uncovering molecular

mechanisms involved in the morphogenesis of larger CNS and in the increased complexity of neuronal connectivity. To evaluate the extent of selective pressure exerted on CDH, this study examined the rate of amino acid and nucleotide substitution between *H. sapiens* and fifteen extant species of primates including strepsirrhines, tarsiers, old and new world monkeys, lesser and great apes covering ~57 MYr of evolution [38], and investigated whether differences in evolutionary constraints are associate with factors known to influence selective pressure including gene locus, essentiality, and expression.

*Abbreviations*: The standard use of human protein (upper case) and gene (upper case italics) symbols are used for all primate species (www.genenames.org/) [39]. When referring to both gene and protein, no italics is used. For mouse proteins and genes, the standard mouse annotation is used (www.informatics.jax.org/mgihome/nomen/gene.shtml#nas).

CD, cytoplasmic domain; CDH; classical cadherins, EC, cadherin domain; ED, ectodomain; FL, full-length; MYr, million years.

## Methods

### Sequence alignment and evolutionary distance analysis

Protein sequences were retrieved from UniProKB (www.uniprot.org) and the National Center for Biotechnology Information (NCBI) (www.ncbi.nlm.nih.gov) databases and the protein topologies described in UniProtKB were used to define the amino acid composition of each domain. Protein coding nucleotide sequences were retrieved from NCBI Genes and Nucleotide databases, and through the University of California Santa Cruz Genome Browser (genome.ucsc.edu) by searching for gene symbol/name and species name. Sequences accession numbers are listed in S2 Table. Type I *CDH15* complete sequence could not be identified in various species of primates and was excluded from this study. The non-human primate species included in this study (listed in S1 Table) were chosen because they cover an evolutionary period of ~57 MYrs, and because their genome assemblies are available. In all cases, variant 1 was chosen if more than one transcript variant was reported. If needed, additional sequence information was obtained from UCSC Genome Browser and using BLAST. The full-length (FL) amino acid sequence of the mature protein (excluding the signal peptide and the cadherin pro-domain) was used for analysis starting at the first amino acid of EC1 and ending at the last C-terminal amino acid of the CD. Sequences were aligned using MUSCLE software within MEGA with default parameters (gap open -2.9; gap extend 0; hydrophobicity 1.2) [40]. Ortholog sequences were aligned and estimates of codon-based evolutionary divergence were conducted in MEGA-X and MEGA-11 [41–43]. Jones-Taylor-Thornton (JTT) matrix-based substitution model was found to have the lowest Bayesian Information Criterion (BIC) and Akaike Information Criterion (AIC) values. To determine the number of amino acid and nucleotide substitutions per site, aligned sequences were analyzed by pairwise comparisons between human and other species using the analytical method, a uniform rate of substitution, and pairwise deletions of gaps following the Kimura-2 model [44].

### Test of selection

The Z-test of selection was used to evaluate the probability of rejecting the null hypothesis of strict neutrality (dN = dS) in favor of negative (dN < dS) or positive (dN > dS) selection. Z-test of selection was conducted in MEGA using aligned FL protein coding nucleotide sequences and the Kumar's method [44]. When the number of nucleotide differences between sequences was small, the Fisher exact test was used to determine whether a sequence was under positive selection.

## Gene essentiality

A score of gene essentiality was computed based on whether the mouse gene knockout was lethal (embryonic or before reproductive age), showed a phenotype affecting reproduction, and whether spontaneous mutations found in humans are associated with severe illness. The data was retrieved from The Jackson Labs databases for mouse phenotypes and human-mouse diseases connections (HMDC) (www.informatics.jax.org). Data were scored as follows: mouse null non-lethal = 0, lethal = 2; mouse phenotype, no phenotype = 0, mild = 1, severe = 2; association with a human disease, none = 0, associated = 1 (maximum essentiality score, 5). The phenotypic MP identification numbers used were: 0002080, embryonic lethality; 0002081, prenatal lethality; 0002082, survival post-natal lethality; 0002083, premature death or induced morbidity; 0002160, abnormal reproductive system morphology; 0001919, abnormal reproductive system physiology [45].

## mRNA abundance

mRNA abundance in human and mouse tissues were obtained from the Genotype Tissue Expression (GTEx) database [46] (gtexportal.org/), which includes 52 human tissues. Single cell mRNA (scRNA) expression levels in human (127 cell types) and mouse (387 cell types) brain samples were obtained from the Allen Institute (https://alleninstitute.org/).

## Statistical analysis

Statistical analyses including the Mann-Whitney U-test, Student's t-test, and Spearman's correlation were conducted in IBM-SPSS Statistics version 29.0.1.0 (IBM Corp, Armonk, NY). Amino acid and nucleotide multiple sequences comparisons were conducted using the Kruskal-Wallis ANOVA analysis followed by Dunn's Test with Holm's procedure for multiple comparisons using R version 4.2.2 (R: A language and environment for statistical computing. R Foundation for Statistical Computing, Vienna, Austria) and R Studio version 2024.04.0 (RStudio: Integrated Development for R. RStudio, PBC, Boston, MA) [47,48]. Non-parametric procedures were used because of the distribution of the dN/dS ratios and outliers noted in many of the comparisons. The Mann-Whitney U test was used for pairwise comparisons of dN/dS ratios between genes located near CDH genes because the data failed a test for normal distribution.

# Results

## Evolutionary changes in CDH protein sequences in primates

To estimate the evolutionary distance between CDH orthologs in various species of primates, pairwise comparisons of amino acid sequences of CDH type I (CDH1, CDH2, CDH3, and CDH4) and type II (CDH5, CDH6, CDH7, CDH8, CDH9, CDH10, CDH11, CDH12, CDH18, CDH19, CDH20, CDH22, and CDH24), and the atypical CDH13, from fifteen extant species were compared to human, representing an evolutionary period of ~57 MYr [38] (S1 Table). The FL protein sequence starting at the first amino acid of EC1 and ending at the last C-terminal amino acid of the CD was used for analysis. Protein sequences were aligned using MUSCLE [40] and distance analysis was conducted by pairwise comparisons between human and each other species using the analytical method, the JTT matrix-based model and a uniform rate of amino acid substitutions in MEGA [41,43].

Fig 1 shows the number of amino acid substitutions per site between human FL type I cadherins and non-human primates plotted over time. CDH1 and CDH3 show a significantly larger number of amino acid substitutions than CDH2 and CDH4, suggesting that CDH2 and CDH4 have been under higher negative selective pressure in primates (Table 1). To evaluate

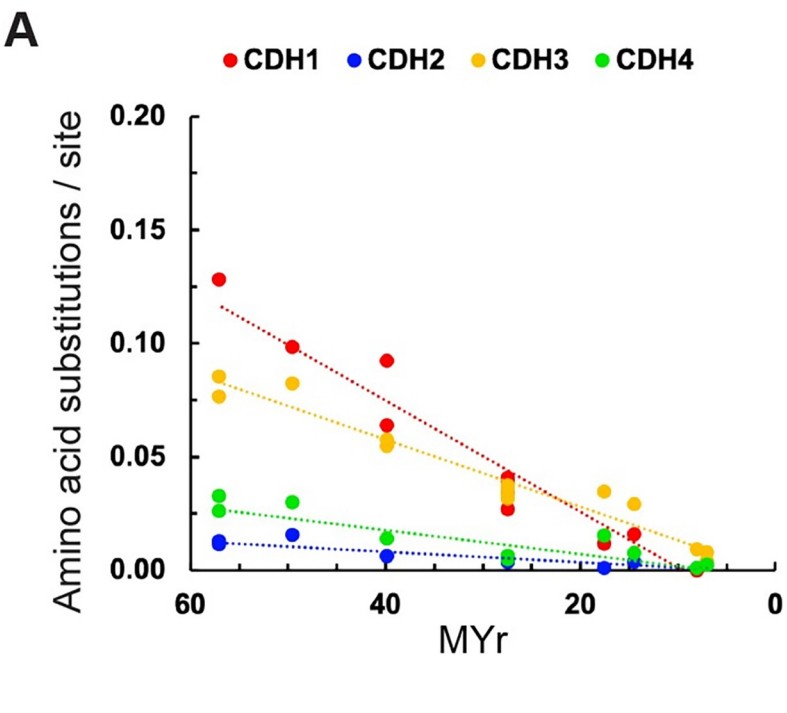

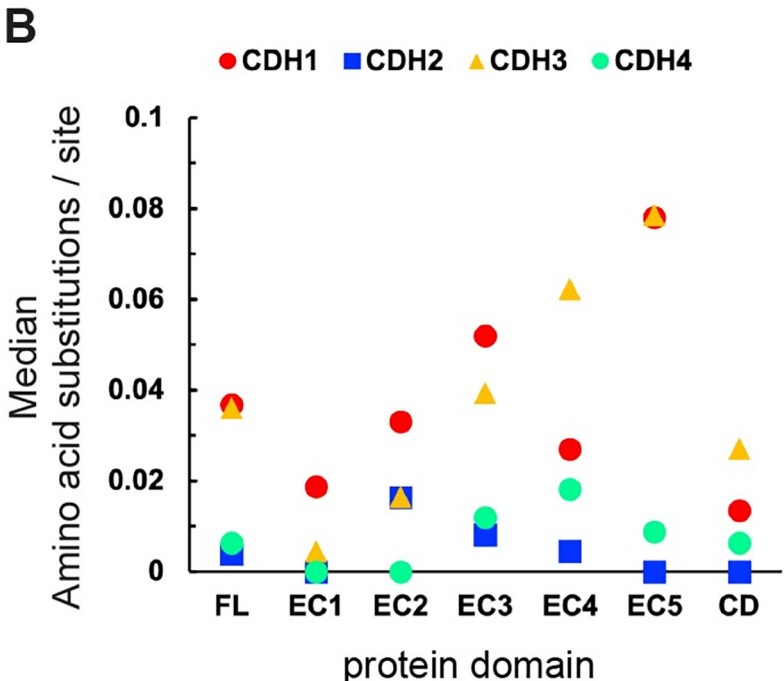

**Fig 1. Amino acid substitutions in type I CDH in primates.** (A) Number of amino acid substitutions per site from pairwise comparisons of aligned FL protein sequences between human and non-human primate vs. time of divergence. Analyses were conducted using the JTT matrix-based model of substitution in MEGA software. Positions containing gaps and missing data were eliminated. Dashed lines represent linear trends using the least squares method. (B) Median number of amino acid substitutions per site in FL, EC1 to EC5, and CD from pairwise comparisons between human and non-human primates. Statistical significance of Kruskal-Wallis ANOVA followed by Dunn's test are shown in Table 1.

the differences in amino acid substitutions along the molecule, the number of substitutions per site in each of the five ECs and the CD were examined by pairwise comparisons (Fig 1B, Table 1). EC1 has the lowest number of substitutions, consistent with its functional relevance in forming high-affinity *trans*-dimer bonds. However, the EC1 of CDH1 shows a significantly higher number of substitutions than CDH2 and CDH4, suggesting that the EC1 of CDH1 has been under less negative selective pressure. EC2 also shows significant differences between CDH1 and CDH4, CDH2 and CDH3, and between CDH3 and CDH4, while the region EC3 to EC5 incorporated a significantly larger number of substitutions in CDH1 and CDH3 as compared to CDH2 and CDH4. The membrane-proximal EC5 is the most distant EC showing

**Table 1. Statistical analysis of amino acid substitutions / site.** Data presented in Figs 1 and 2.

| | FL | EC1 | EC2 | EC3 | EC4 | EC5 | CD |
|---|---|---|---|---|---|---|---|
| **CDH type I** | | | | | | | |
| Fig 1 | | | | | | | |
| CDH1 vs CDH2 | *** | ** | ns | ** | * | ** | ** |
| CDH1 vs CDH3 | ns | ns | ns | ns | ns | ns | ns |
| CDH1 vs CDH4 | * | * | *** | ** | ns | * | ns |
| CDH2 vs CDH3 | *** | ns | * | ** | *** | *** | *** |
| CDH2 vs CDH4 | ns | ns | ns | ns | ns | ns | * |
| CDH3 vs CDH4 | ** | ns | *** | ** | * | ** | * |
| **CDH type II** | | | | | | | |
| Fig 2A and 2E | | | | | | | |
| CDH10 vs CDH6 | ** | ns | ns | *** | ns | ns | ** |
| CDH10 vs CDH9 | * | ns | ns | ns | ** | * | ns |
| CDH6 vs CDH9 | ns | ns | ns | ns | ns | ns | ns |
| Fig 2B and 2F | | | | | | | |
| CDH12 vs CDH18 | ns | *** | ns | ns | *** | ns | ns |
| CDH12 vs CDH20 | ns | *** | ns | ns | ns | ns | ns |
| CDH12 vs CDH22-FL | ns | *** | ns | *** | ns | ns | ns |
| CDH12 vs CDH7-FL | ns | *** | ns | ns | ns | ns | ** |
| CDH18-FL CDH20-FL | ns | ns | ns | ns | ** | ns | ns |
| CDH18 vs CDH22-FL | ns | ns | * | ** | ** | ns | ns |
| CDH18 vs CDH7-FL | *** | ns | 1 | ns | *** | ns | ns |
| CDH20 vs CDH22 | ns | ns | ns | ** | ns | ns | ns |
| CDH20 s CDH7 | ns | ns | ns | ns | ns | ns | ns |
| CDH22 vs CDH7-FL | ns | ns | * | * | ns | ns | * |
| Fig 2C and 2G | | | | | | | |
| CDH11 vs CDH24 | * | ns | ns | ns | ** | *** | *** |
| CDH11 vs CDH8-FL | * | ns | ns | *** | * | ns | ns |
| CDH24 vs CDH8-FL | ** | ns | ns | *** | *** | *** | *** |
| Fig 2D and 2H | | | | | | | |
| CDH13 vs CDH19-FL | ** | *** | ** | *** | ** | ** | |
| CDH13 vs CDH5-FL | ns | ns | *** | *** | ns | *** | |
| CDH19 vs CDH5-FL | ns | ns | ns | ns | ** | ns | ns |

Kruskal-Wallis ANOVA followed by Dunn's test p adjusted:

*p ≤ 0.05;

**p ≤ 0.01;

*** p≤ 0.001;

ns, not significant.

the least homology among ECs; however, a significantly lower number of amino acid substitutions is observed in the EC5 of CDH2 and CDH4 as compared to CDH1 and CDH3, suggesting that EC5 has been under higher negative selection in CDH2 and CDH4, possibly due to a functional role of this domain in these cadherins (see discussion). The amino acid sequence similarity and topology of the CD is the defining feature of CDH and has varied slightly among primates. However, the CD of CDH2 shows a significantly lower number of amino acid substitutions as compared to the other type I CDH, suggesting that CDH2 has been under higher evolutionary constraints. This analysis shows that CDH type I have been under high negative selective pressure in primates; however, a significantly higher number of substitutions is observed in CDH1 EC1, and a high degree of variation is found in the EC3 to EC5 region of the proteins.

CDH type II are grouped based on their heterophilic binding affinity: group A (CDH6, CDH9, and CDH10), group B (CDH7, CDH12, CDH18, CDH20, and CDH22), group C (CDH8, CDH11, and CDH24), and ungrouped (CDH5 and CDH19) [24]. Pairwise comparisons of FL protein sequences between human and non-human primates were conducted within the three specificity groups. The more distant CDH5 and CDH19, and the atypical CDH13 were analyzed as ungrouped CDH (Fig 2A–2D, and Table 1), CDH type II in groups A, B, and C, show very few amino acid substitutions over time, indicating that these cadherins have been under high evolutionary constraints in primates. Within group A, CDH10 has changed significantly less than CDH6 and CDH9. Analysis of the number of amino acid substitutions per site within each protein domain shows no significant differences in EC1 and EC2, while comparisons of the EC3 to EC5 and the CD of CDH10 show a significantly lower number of amino acid substitutions. Analysis of FL group B CDH shows no significant differences except between CDH7 and CDH18. However, statistical comparisons of each protein domain show significant differences in EC1 between CDH12 and the other group B members. A few significant differences are observed in EC domains between CDH22 and the other group members, and no differences are observed in EC5 and CD within group B cadherins. Group C CDH type II have also changed slightly over time. While no detectable differences are observed in EC1 and EC2, the protein region from EC3 to EC5 and the CD has changed more significantly in CDH24 as compared to CDH8 and CDH11, suggesting that these two protein regions may have different functional roles among these cadherins. The ungrouped CDH5 and CDH19 have varied more extensively than the grouped type II cadherins, having rates of amino acid substitutions similar to those observed in CDH1 (Fig 2D). No significant differences are observed between CDH5 and CDH19 in most protein domains; however, both show significant differences with CDH13, which has a rate of amino acid substitution similar to the grouped CDH type II. In summary, while all grouped CDH type II and the atypical CDH13 show similar rates of amino acid substitutions as the ones observed in CDH2, the differences in the number of substitutions of CDH5 and CDH19 are similar to the ones observed in CDH1. The average rate of amino acid substitutions / site / MYr has been > 10 times higher in CDH1, CDH3, CDH5, and CDH19 over the 57 MYr analyzed (CDH1, 0.12%; CDH3, 0.14%; CDH5, 0.12%; CDH19, 0.15%) as compared to < 0.01% average rate of amino acid substitutions in all other CDH. This analysis shows that negative selective pressure has been high among CDH in primates; however, the subset of cadherins, including CDH1, CDH3, CDH5 and CDH19 have changed more than the other family members.

## Tests of selection

Nucleotide substitutions within protein coding sequences are expected to occur randomly. Because synonymous substitutions are considered neutral for protein function, they

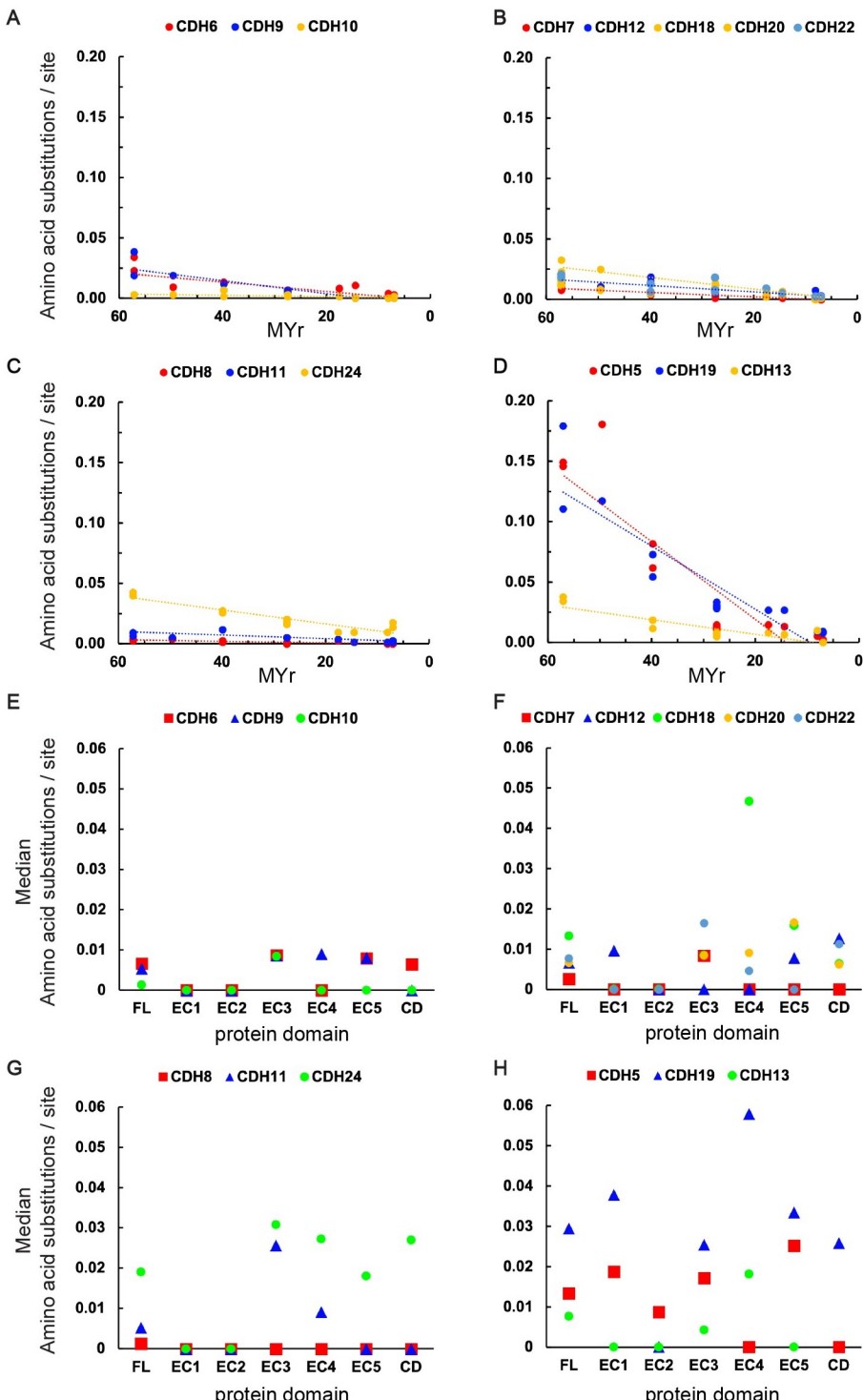

**Fig 2. Amino acid substitutions in CDH type II in primates.** (A to D) Number of amino acid substitutions per site of aligned FL protein sequences between human and non-human primates vs. divergence. Analyses were conducted using the JTT matrix-based model in MEGA software. Positions containing gaps and missing data were eliminated. Dashed lines represent linear trends calculated by the least squares method. (E to H) Median number of amino acid substitutions per site in FL, EC1 to EC5, and CD protein sequence. Kruskal-Wallis ANOVA of pairwise comparisons followed by Dunn's test were conducted within each specificity group (A, B, and C) and ungrouped cadherins. Statistical significances (p adjusted) of pairwise comparisons followed by Dunn's test are shown in Table 2.

accumulate regularly over time. In contrast, non-synonymous substitutions may be under selective pressure depending on the effect of the amino acid change on protein function. Therefore, the ratio between the number of non-synonymous substitutions per non-synonymous sites (dN) over synonymous per synonymous sites (dS) between aligned protein coding nucleotide sequences of ortholog genes is commonly used to estimate the strength of positive and negative selection [49,50]. A dN/dS ratio of 1 indicates neutral selection, while dN/dS ratios > 1 and < 1 are used as a metric of positive and negative selection respectively. However, because amino acid substitutions are most commonly deleterious, dN/dS values in most protein coding sequences are < 1, and dN/dS ratios provide a conservative estimate of positive selection [50].

To determine whether CDH have been under neutral, negative, or positive selection in primates the Z-test of selection and the Fisher's exact test of selection for small samples were conducted in MEGA. Aligned FL protein coding nucleotide sequences and Kumar's method were used to evaluate the probability of rejecting the null hypothesis of strict neutrality in favor of negative or positive selection [44]. Both the Z-test and Fisher's exact test of selection rejected positive selection for CDH type I and II (p > 0.05). CDH comparisons between humans and non-human primates show that in most cases the null hypothesis of strict neutrality could be rejected in favor of the alternative hypothesis of negative selection (p < 0.05) (S3 Table). In a few cases pairwise comparisons between human and one species show that strict neutrality could not be rejected in favor of negative selection, (*CDH3 Pongo abelii, CDH10 Pan paniscus,* and *CDH24 Pan troglodytes*). Z-test analysis of *CDH19* shows that strict neutrality could not be rejected between humans and apes, indicating that this gene has been under lower negative selection and that adaptative changes may have occurred. Therefore, the tests of selection indicate that except for *CDH19,* both CDH type I and II have been under negative or purifying selection.

## Pairwise comparisons of nucleotide substitutions between CDH

To examine the evolutionary changes among protein coding nucleotide sequences of CDH orthologs, the number of nucleotide substitutions per site between FL protein coding sequences was estimated by pairwise comparisons of aligned sequences between humans and non-human primates using the Kimura's 2-parameter model of substitutions in MEGA. CDH type I dS values show a consistent decline in distance between species over time and no statistically significant differences are observed (p > 0.05) (Fig 3, Table 2). In contrast, a comparison of dN and dN/dS values between human and non-human primates shows significantly higher values in *CDH1* and *CDH3* as compared to *CDH2* and *CDH4* (Fig 3B, Table 2). Fig 3C shows the average dN/dS of three groups of primates, which diverged from humans >39 MYr ago (new world monkeys, *tarsiers* and *strepsirrhines*), ~27 MYr ago (old world monkeys), and ~10 MYr ago (apes). A pronounced decrease in *CDH1* dN/dS ratios is observed between 27 and 10 MYr, while dN/dS values of *CDH3* have remained consistently higher. *CDH2* and *CDH4* show consistently low dN/dS ratios throughout the 57 MYr studied, indicating that high negative selection on these CDH predates primates. The number of changes in *CDH1* and *CDH3* have been more frequent in the ED relative to the CD, while both domains of *CDH2* and *CDH4* have remained equally unchanged (Fig 3D). However, the CD of *CDH2* shows significantly lower dN values when compared to *CDH4,* indicating that this domain has been under higher negative selection in *CDH2.*

Pairwise comparisons between human and non-human primates of CDH type II orthologs were used to estimate dS and dN values and plotted vs. time of divergence (Fig 4A to 4H). Statistical comparisons of each specificity group show no significant differences in dS values

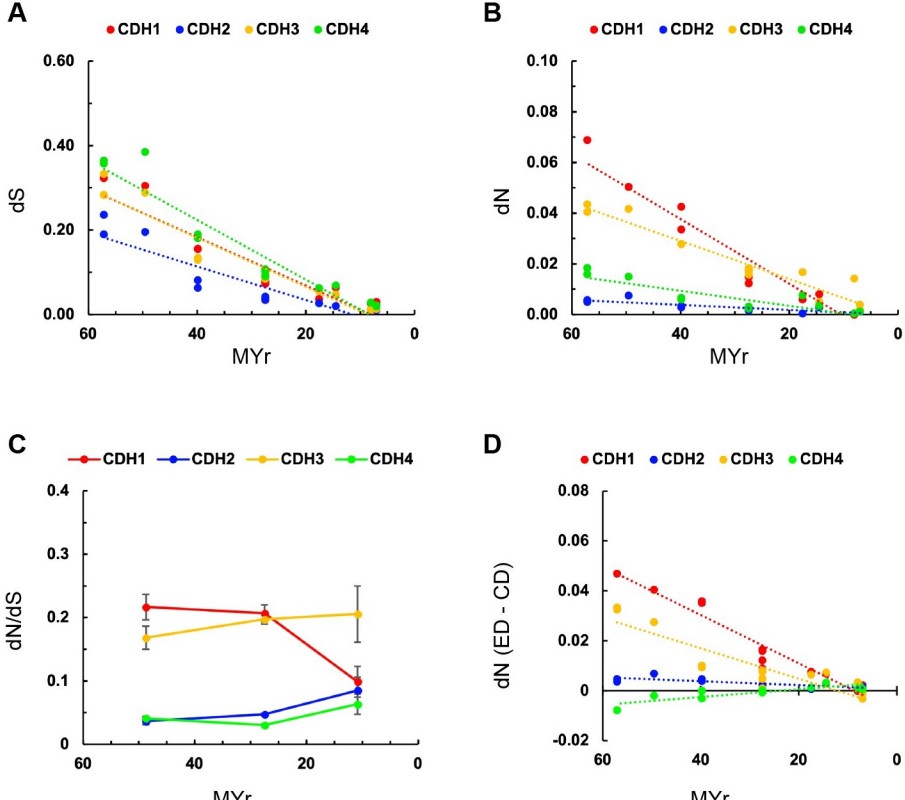

**Fig 3. Estimates of evolutionary divergence between human and non-human primates in CDH type I protein coding nucleotide sequences.** (A, B) Number of synonymous substitutions per synonymous site (dS) and non-synonymous substitutions per non-synonymous site (dN) from pairwise comparisons between human and non-human primate orthologs vs. time of divergence. (C) Comparisons of dN/dS ratios between humans and non-human primates. The average dN/dS values of non-human primates grouped by their time of divergence (apes 10.9 MYr, New World monkeys 27.5 MYr, and old world monkeys, tarsiers, and Strepsirhines 48.7 MYr). dS and dN values were calculated using Kumar's method [44], Kimura 2-parameters model, and the analytical formula with a uniform rate of substitutions and pairwise deletions of ambiguous positions in MEGA [41,43]. Datapoints are the average of each group and error bars represent the standard error of the mean. (D) Difference in dN values between the ED and CD of each CDH plotted over time. Kruskal-Wallis ANOVA of pairwise comparisons followed by Dunn's test are shown in Table 2. The averaged dN/dS of each group of primates were only used for plotting, while dN/dS values between humans and each primate species were used for statistical analysis of the data shown in panel C.

(Fig 4A to 4D, Table 2). In contrast, ungrouped *CDH5* and *CDH19* show higher dN values, which are similar to the ones observed in *CDH1* (Fig 4E to 4H). Within group A, *CDH10* shows significantly lower dN and dN/dS values than *CDH6* and *CDH9*, suggesting that *CDH10* has been under higher negative selective pressure. Comparisons of dN values between ED and CD show no significant differences between the two regions of the protein (Fig 4M). No significant differences in dN values are observed among group B cadherins except for the comparison between *CDH7* and *CDH18*. *CDH7* also has dN/dS ratios significantly lower than *CDH12*, *CDH18*, and *CDH22* (Fig 4B, 4F and 4J, Table 2). *CDH12* and *CDH22* show higher dN values in the CD relative to the ED, suggesting that the ED of this group has been under higher evolutionary constraints as compared to the CD (Fig 4N). Within group C cadherins, *CDH24* shows the largest difference in dN and dN/dS values between humans and the *Microcebus / Otolemur* group as compared to *CDH8* and *CDH11* (Fig 4C and 4G). *CDH24* shows the largest difference in dN and dN/dS values between humans and non-human primates, suggesting that *CDH24*

**Table 2. Statistical analysis of CHD dS, dN, and dN/dS.** Data presented in Figs 3 and 4.

| | dS | dN | dN/dS | dN (ED—CD) |
|---|---|---|---|---|
| **CDH type I** (Fig 3) | | | | |
| CDH1 vs CDH2 | ns | *** | *** | * |
| CDH1 vs CDH3 | ns | ns | ns | ns |
| CDH1 vs CDH4 | ns | ns | *** | *** |
| CDH2 vs CDH3 | ns | *** | *** | ns |
| CDH2 vs CDH4 | ns | ns | ns | * |
| CDH3 vs CDH4 | ns | * | *** | *** |
| **CDH type II** (Fig 4) | | | | |
| **Group A** | | | | |
| CDH10 vs CDH6 | ns | *** | *** | ns |
| CDH10 vs CDH9 | ns | * | * | ns |
| CDH6 vs CDH9 | ns | ns | * | ns |
| **Group B** | | | | |
| CDH12 vs CDH18 | ns | ns | ns | ** |
| CDH12 vs CDH20 | ns | ns | ns | ns |
| CDH12 vs CDH22 | ns | ns | ns | ns |
| CDH12 vs CDH7 | ns | ns | * | * |
| CDH18 vs CDH20 | ns | ns | ** | ns |
| CDH18 vs CDH22 | ns | ns | ns | * |
| CDH18 vs CDH7 | ns | * | *** | ns |
| CDH20 vs CDH22 | ns | ns | ns | ns |
| CDH20 vs CDH7 | ns | ns | ns | ns |
| CDH22 vs CDH7 | ns | ns | *** | * |
| **Group C** | | | | |
| CDH11 vs CDH24 | ns | *** | ** | *** |
| CDH11 vs CDH8 | ns | * | ** | ns |
| CDH24 vs CDH8 | ns | *** | *** | ns |
| **Ungrouped** | | | | |
| CDH13 vs CDH19 | ns | *** | *** | |
| CDH13 vs CDH5 | ns | * | ** | |
| CDH19 vs CDH5 | ns | ns | * | ns |

Kruskal-Wallis ANOVA followed by Dunn's test, p. adjusted.

* $p \leq 0.05$;

** $p \leq 0.01$;

*** $p \leq 0.001$;

ns, not significant.

has been under significantly lower negative selective pressure (Fig 4K and 4O, Table 2).
Ungrouped *CDH5* and *CDH19* show the largest difference in dN values between humans and
non-human primates within type II cadherins, and both cadherins are significantly different
from the atypical *CDH13* (Fig 4D and 4H). *CDH5* is the only CDH type II showing relatively
higher dN values in the ED than the CD (Fig 4P). *CDH19* dN/dS ratios are doubled between
apes and humans and show the largest difference among all species, suggesting that adaptative
changes may have occurred in this molecule during relatively recent evolution (Fig 4L). dN/dS
ratios in *CDH13* are similar to the values observed in grouped type II cadherins and *CDH2*
(Fig 4L and 4P). This analysis indicates that CDH have been under high negative selective

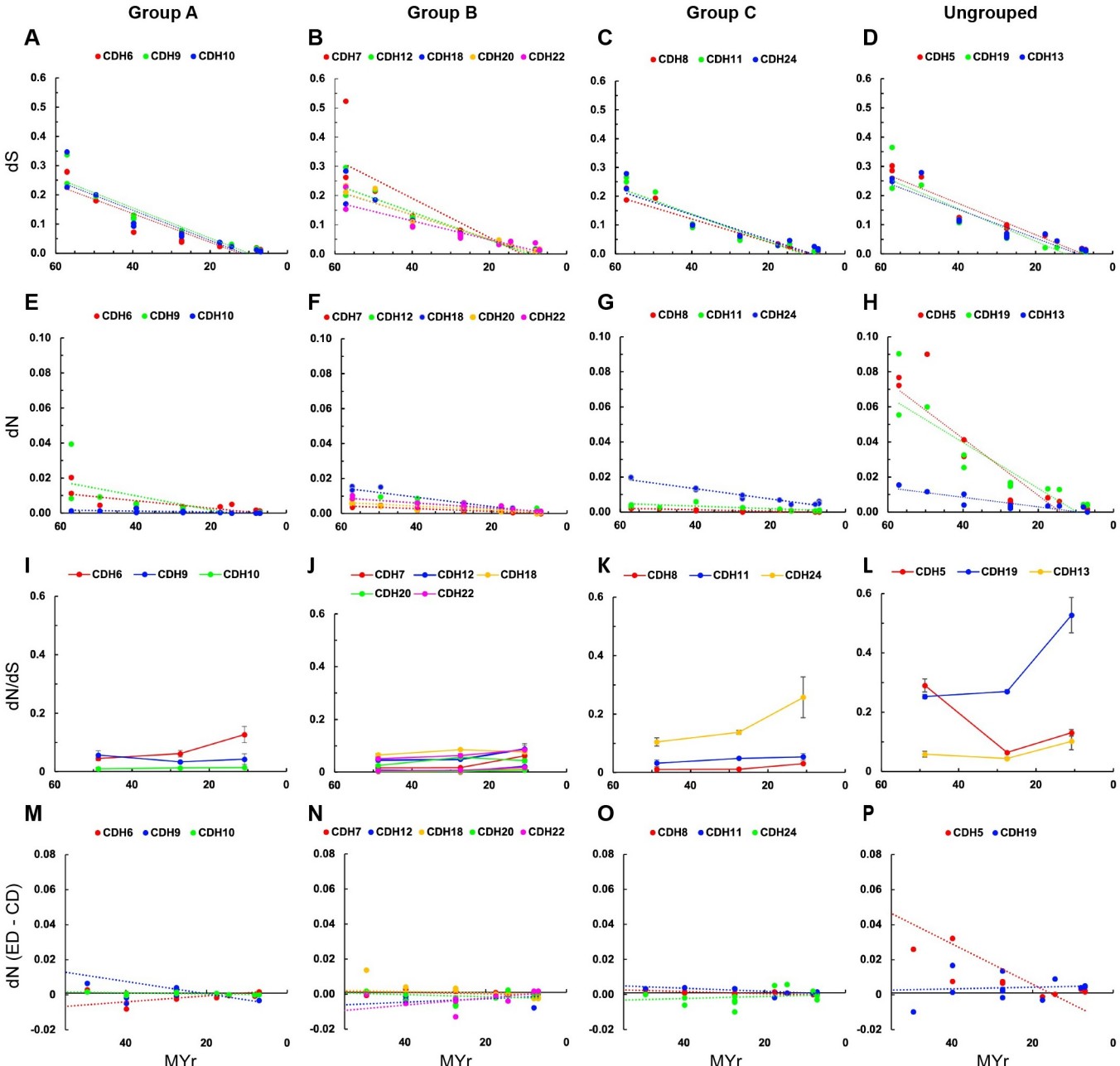

**Fig 4. Estimates of evolutionary divergence between human and non-human primates in type II cadherins protein coding nucleotide sequences.** Aligned protein coding nucleotide sequences were used to estimate the dS and dN values between human and non-human primates orthologs. dS and dN values were calculated using Kumar's method [44], Kimura 2-parameters model, and analytical formula using the uniform rate of substitutions and pairwise deletions in MEGA. (A to H) dS and dN values over time of specificity group A (A, E), group B (B, F), group C (C, G) and ungrouped (D, H). (I to L) Comparisons of dN/dS ratios between human and non-human primates within groups A, B, C and ungrouped respectively. The average of dN/dS values of non-human primates (apes 10.9 MYr, new world monkeys 27.5 MYr, and old world monkeys, tarsiers, and strepsirrhines 48.7 MYr) are shown. (N to P) Difference of dN values between the ED and CD of each cadherin are plotted over time. Dashed lines represent linear trends using the least squares method. Statistical significance (p adjusted) Kruskal-Wallis ANOVA of pairwise comparisons followed by Dunn's test are shown in Table 2.

pressure in primates; however, significant differences are observed among them. While type I *CDH2* and *CDH4*, and grouped type II cadherins have varied slightly since the time of divergence from the human lineage, dN and dN/dS values of *CDH1*, *CDH3*, *CDH5*, and *CDH19* are significantly larger as compared to the other cadherins. These higher dN and dN/dS may be associated with adaptative changes, lower relevance for fitness, or different expression patterns. Previous studies showed that coding sequences of N-glycosylated proteins expressed on the cell surface evolve more rapidly than the ones localized to the cytosol [45,51,52]. Therefore, within CDH, the rate of change observed in *CDH1*, *CDH3*, *CDH5*, and *CDH19* agrees with these previous findings, while most CDH tend to differ from these observations.

## CDH rate of evolution and chromosomal location

To examine whether the dN/dS ratios observed in CDH genes are associated with their locus, genes coding for transmembrane proteins located in the proximity of each cadherin gene were identified, and the dN and dS values between human and non-human primates orthologs were calculated in MEGA. Fig 5A and 5B show dN/dS ratios in box and whisker plots of type I and type II cadherins and genes located within the same chromosomal region, respectively. Pairwise comparisons of dN/dS ratios between each cadherin and a near gene were conducted by Mann-Whitney U-test in SPSS (*p < 0.001). In all cases, the dN/dS ratio of the near gene is similar or higher than the ratio of the CDH gene, and in most cases, a statistically significant difference was detected. This suggest that the high level of negative selection exerted over type I and II cadherins is not related to their chromosomal location.

## Correlation between gene essentiality and CDH evolutionary distance

Several factors correlate with the rate of protein evolution, including, the essentiality of the gene for survival and reproductive fitness, gene expression levels, and subcellular localization [45,53]. To investigate whether the essentiality of CDH is associated with differences in the rate of evolution, an essentiality score for each cadherin was calculated and compared with the evolutionary distance between human and mouse orthologs. A gene essentiality score was computed based on whether the mouse gene knockout was lethal (embryonic and before reproductive age), showed a phenotype affecting breeding, and whether spontaneous mutations in humans have been associated with a severe disease (see Methods). Spearman's correlation coefficients of 0.383 and 0.302 were found between the essentiality scores and dN and dN/dS values, respectively (p > 0.05, 2-tailed), indicating a low, non-significant correlation between essentiality scores and CDH evolutionary distance. These results suggest that factors not related to the parameters commonly used to determine gene essentiality have influenced the rate of CDH change or that the commonly used criteria for scoring gene essentiality may not appropriately reflect the role of these genes on the fitness of the organisms.

## Correlation between CDH expression levels and rate of evolution

A negative correlation between gene expression levels and the rate of protein evolution has been consistently observed in a variety of genes and species, indicating that highly expressed genes are under higher evolutionary constraints than lowly expressed genes. This observation is the expression level—evolutionary rate (E-R) anti-correlation [54–59]. Gene expression levels may reflect the functional relevance of a gene for the fitness of the organisms, and therefore, highly expressed genes are under higher evolutionary constraints. However, the reasons for the E-R anticorrelation between ortholog genes are not entirely understood, and several hypotheses have been postulated to explain it [60–62]. In contrast to the typically observed E-R anti-correlation, some gene families show a positive E-R correlation, which is associated

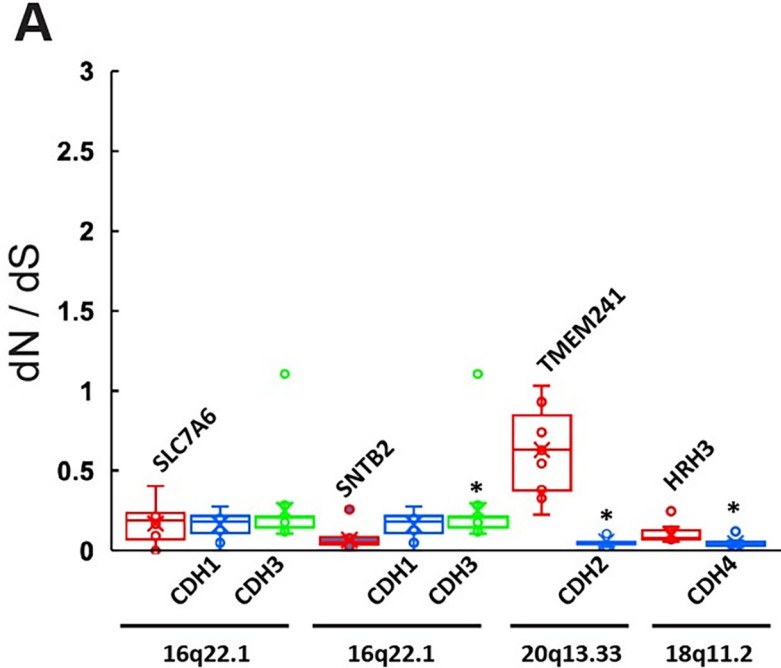

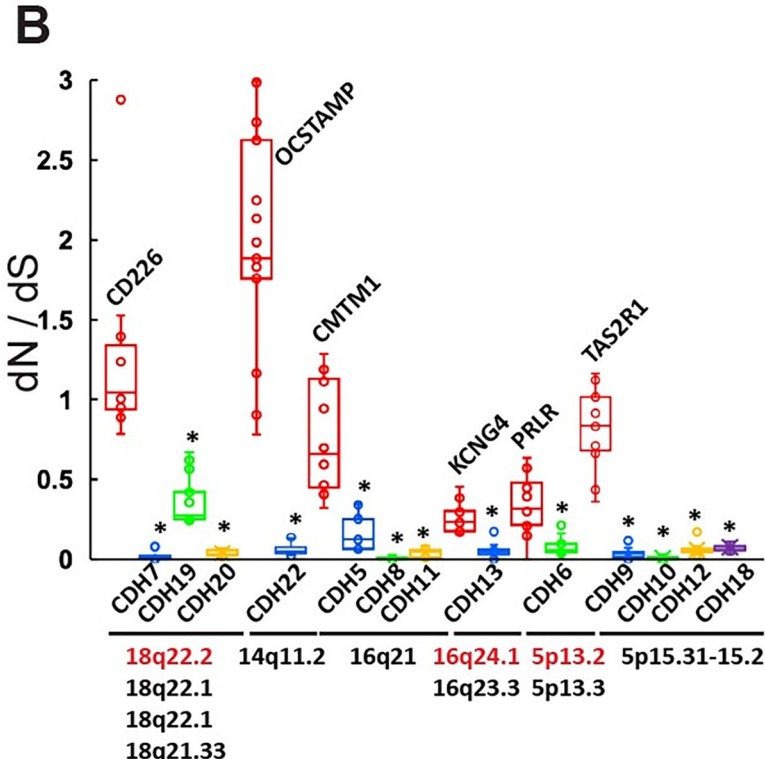

**Fig 5. Comparison of dN/dS ratios between CDH and transmembrane protein coding genes located in the same chromosomal region.** dN/dS ratios of protein coding nucleotide sequences between human and non-human primates were calculated and plotted as box and whiskers with the median indicated as a line within the box. dN/dS ratios of each non-cadherin gene are plotted in red, and their chromosomal location is shown below the *x*-axis. Comparisons of dN/dS ratios between each (A) type I and (B) type II cadherin with the closest transmembrane protein coding gene,

respectively. In the case the chromosomal location of the closet gene differs from the cadherin gene, their locations are displayed in red and in black letters, respectively from top to bottom in the order of the genes shown within the plot from left to right. * Mann-Whitney U-test pairwise comparisons between each cadherin and the gene located in the same chromosomal region with an exact significance < 0.001 (2-sided test).

with the subcellular localization of the protein (cytoplasmic vs. extracellular) and with protein glycosylation. N-glycosylated extracellular proteins tend to have a positive E-R correlation, which may be associated with the quality control mechanisms regulating cell surface expression and exocytosis of properly folded proteins [45,51,52]. Because CDH are N-glycosylated transmembrane proteins expressing most of the polypeptide on the cell surface, a positive E-R correlation is expected to be observed. Spearman's correlation analyses of dN and dN/dS ratios between human and non-human primates vs. cadherins expression levels in 52 human tissues (Genotype Tissue Expression (GTEx) [46]) show a significant E-R positive correlation with their expression level in most tissues (Fig 6A). These results are agreement with previous studies indicating that the evolutionary rate of extracellularly expressed N-glycosylated proteins positively correlates with their level of expression [51,52]. However, a significant E-R anti-correlation was observed with CDH expression in the human CNS (Fig 6A, S1 File), indicating that positive and negative E-R correlations of CDH vary depending on their expression pattern.

To examine whether the expression levels of CDH in the CNS correlate with their evolutionary rates, cadherins dN and dN/dS values were compared with their expression levels in single human and mouse CNS cells estimated by scRNA expression levels available at the Allen Institute database [63,64]. From the 127 human cell types analyzed (109 neurons and 18 non-neuronal cells) a significant E-R anti-correlation was observed with at least one species of primates in 100% of the neurons, while 27% of non-neuronal cells show a significant negative E-R correlation (Spearman's correlation p < 0.05, 2-tailed) (Fig 6B, S1 File). Similarly, from a total of 387 cell types analyzed from the mouse cerebral cortex and hippocampal formation (364 neurons and 23 non-neuronal cells), 261 (67.44%) cells (255 neurons and 6 non-neuronal cells) show a significant negative E-R correlation (Spearman's p < 0.05, 2-tailed) (Fig 6C, S1 File). These results show that in 70.1% of the neurons analyzed, a significant E-R anti-correlation with evolutionary distance is observed between human and mouse. In comparison, 26.1% of non-neuronal cells show a significant E-R anticorrelation. In summary, Spearman's correlation analysis shows a significant positive E-R correlation when comparing CDH evolutionary distances with their expression levels in most human tissues, in agreement with previous studies indicating that extracellular N-glycosylated proteins have a higher rate of change when compared with intracellular un-glycosylated proteins [51,52]. In contrast, a significant E-R anti-correlation is observed between the rate of cadherins evolution and their expression levels in CNS tissues and neurons. This suggests that the differences in evolutionary constraints observed among CDH are associated with their expression and role in CNS development and function.

Analysis of genetic variants predicted to inactivate gene function in humans is used to estimate the functional relevance of protein coding genes in human physiology [65]. The Genome Aggregation Database (gnomAD, https://gnomad.broadinstitute.org) is a comprehensive compilation of genetic data from human exomes and genomes. Using this dataset, a high-confidence metric of intolerance to predicted loss-of-function (pLoF) variants in humans was developed based on the assumption that gene variants with phenotypic consequence affecting human health are expected to be less frequent in the population as compared to variants with no deleterious effect [65]. The LOEUF score represents the upper bound fraction of the ratio

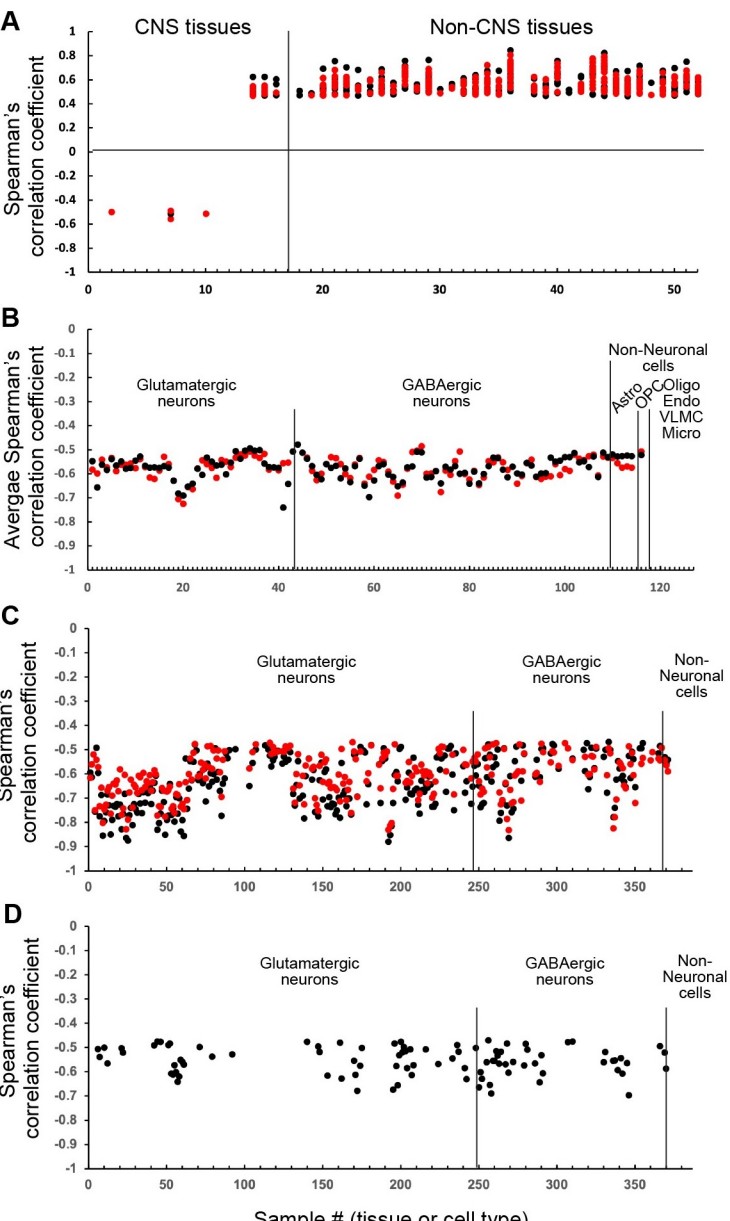

**Fig 6. Spearman's correlation analysis of type I and type II cadherins dN and dN/dS values between human and non-human primates vs. cadherins mRNA expression levels in human and mouse tissues.** (A) Correlation of dN and dN/dS values vs. cadherins expression in 52 human tissues (GTEx database). Samples 1–13, CNS structures; 14–52 non-CNS tissues (see S1 File for details). Statistically significant (p < 0.05) 2-tailed correlation values are plotted (dN, red dots; dN/dS, black dots). (B) Correlation analysis of cadherins dN and dN/dS values vs. cadherins scRNA expression in 127 cell types from human cerebral cortex obtained from the Allen Institute cell atlas. The average of the statistically significant (p < 0.05) 2-tailed correlation values of dN (red dots) and dN/dS (black dots) for each cell type are plotted. Samples: 1–109, neurons; 110–127, non-neuronal cells; 1–42, glutamatergic neurons; 43–109, GABAergic neurons; 110–114, astrocytes; 115–116, OPC; 117–120, oligodendrocytes; 121–123, endothelial cells; 124–125, VLMC; 126–127, Micro-PVM. (See S1 File for details). (C) Statistically significant (p < 0.05; 2-tailed) Spearman's correlation values of cadherins dN (red dots) and dN/dS black dots), and (D) LOEUF scores vs. cadherins scRNA expression in 387 cell types from mouse cerebral cortex obtained from Allen Institute cell atlas. Samples #: 1–242, glutamatergic neurons (1–130, isocortex; 131–238 hippocampus; 239–242, unique); 243–364, GABAergic neurons (243–329, isocortex; 330–352, hippocampus; 353–364, other); 365–383, non-neuronal cells (see S1 File for details).

between observed loss-of-function variants / expected number of variants in individual protein coding genes. It is used to identify the degree of intolerance to pLoF variants and their relevance for human physiology [65]. A lower than expected number of observed variants in a gene (low LOEUF score), suggests that certain variants have been depleted from the human population, indicating higher evolutionary constraints presumably due to the functional relevance of the gene. A Spearman's correlation analysis of CDH LOUEF scores and dN/dS values between humans and new world monkeys shows a significant positive correlation (p<0.05, 2-tailed) (S4 Table), indicating that the higher negative selection observed during primate evolution exerted on CDH is reflected on the expected relevance of the genes in human physiology based on their pLoF. Spearman's correlation analysis between CDHs LOEUF scores and scRNA expression levels in 387 mouse brain cells shows a statistically significant negative correlation (p<0.05, 2-tailed) in 24% of the cells (90 neurons and 3 non-neuronal cells) (Fig 6D). These analyses agree with the above presented data showing that CDH highly expressed in the CNS have been under higher evolutionary constraints in primates including humans presumably due to their role in CNS development and function.

## Discussion

The emergence of cell-cell adhesion and communication played a central role in the integration of cells into tissues and systems performing the complex biological functions observed in multicellular organisms. The advent of the cadherin family of cell adhesion and signaling molecules coincides with the appearance of multicellularity and is thought to derive from ancestral multi-domain proteins that evolved through protein motifs deletions and duplication and the addition of a CD ([4,12,66] and references within). The first expansion of cadherins aligns with the onset of metazoans ~600 MYr ago, when a single gene found in protozoans expanded to ~30 genes in choanoflagellates [6,67]. The later diversification into the present-day superfamily occurred in vertebrates, in which ~119 genes are found [4,6,12,68].

The spatial and temporal expression of CDH is highly regulated, displaying distinct patterns throughout the body [32,69,70]. During neural development, CDH participate in diverse morphogenic processes including asymmetric cell division, cell migration, neural tube segmentation, neurite outgrowth and pathfinding, axonal branching, and synapse formation [1,34,71–74]. CDH1 and CDH2 are expressed in epithelial and neuroectodermal cells respectively, and germ-line deletions and function-blocking perturbations cause tissue malformations and embryonic lethality [75,76]. Due to their essentiality in animal development, both genes are expected to be under similar negative selective pressure. However, soon after their discovery, it was noted that cadherins orthologs predominantly expressed in the CNS show higher homology than those expressed in non-neural tissues [37]. In primates, type I cadherins expressed in neurons are phylogenetically closer than those expressed in non-neuronal cells, while the degree of homology between species varies among cadherin domains with the highest homology observed in EC1-EC2. Considering the importance of the EC1 in *trans*-dimer formation, this domain is expected to be under high negative selective pressure. However, the EC1 of *CDH1* incorporated more non-synonymous substitutions than any other CDH, which may have affected its homophilic binding affinity. In fact, the homodimerization Kd of human CDH1 is ~10 times higher than CDH2 (217.0 μM and 22.1 μM respectively), and it is higher than the Kd found in chicken (110 μM) and mouse (160 μM), which diverged from humans ~320 and ~80 MYr ago respectively. In contrast, the homodimerization Kd of CDH2 remained constant during the same period (20.4 μM in chicken, 22.6 μM in mouse and 22.1 μM in human) [26]. Cadherin's function in epithelia appears to be less dependent on their homophilic binding affinity, while their role in neural

tissue highly relies on their homo/heterophilic binding specificity. Even though CDH1 and CDH2 have similar size, topology, processing, subcellular localization, and share ~70% of sequence similarity [3], they show significantly different rates of non-synonymous substitutions, suggesting that differences in expression between neural and non-neural tissues have been an important factor in their evolution.

*CDH1* and *CDH3* are closely located in human chromosome 16 and are expressed in epithelia, including skin, digestive, and reproductive systems. *Cdh1* null mice die before birth and *Cdh3* null mice develop a defective reproductive system; therefore, their essentiality is expected to be similar for both genes [76–78]. Indeed, no significant differences in dN/dS ratios were found between *CDH1* and *CDH3* ED in primates. However, a significantly larger number of non-synonymous substitutions is observed in the CD of *CDH3* as compared to *CDH1*, suggesting that regulation of cytoskeletal dynamics and/or signaling has been a driving force in *CDH3* evolution. In contrast to *CDH1*, *CDH2* and *CDH4* have remained largely unchanged in primates, indicating that both genes have been under similarly higher negative selective pressure. In the CNS, CDH2 is found at synapses and participates in pre and postsynaptic differentiation, synaptic transmission, and plasticity [79–88]. Germ-line deletion of *Cdh2* in mice causes death before reproductive age and ectopic expression of Cdh2 disrupts tissue morphogenesis [75,89,90]. *Cdh4* is also predominantly expressed in the mouse CNS but at lower levels than *Cdh2*. Null *Cdh4* mice are viable and, therefore, have lower essentiality scores than *Cdh2* [91]. The high level of expression of both genes in neural tissues and the negative correlation observed between dN and dN/dS values vs. cadherins expression in the CNS suggest that their role in neural development and function is relevant for the fitness of the organism and has been a determinant factor in their evolution.

The EC5 of CDH2 and CDH4 has been under significantly higher negative selective pressure in primates as compared to CDH1 (Fig 1), suggesting that EC5 has played a distinct function during CNS evolution. The folding of EC5 differs from the typical EC and it is the least homologous domain among CDH [3]. Structure-function analysis of cadherins binding *in vitro* showed that EC4-EC5 do not form *trans*-dimers but contribute to *trans*-dimerization between EC1-EC2 [92], possibly by maintaining an extended ED rod of the required size. *In vivo* analysis of the mouse cerebral cortex development showed that CDH2 is involved in the regulation of Fibroblast Growth Factor Receptor (FGFR) expression levels on the cell surface. FGF signaling regulates neuronal orientation, migration, and positioning within the mammalian cortex, and therefore, changes of FGFR expression levels affect the laminar organization of the cerebral cortex [93]. Analysis of CDH *cis*-interactions with FGFR showed that both, CDH1 and CDH2 interact with FGFR via EC4-EC5; however, only CDH2 is capable of maintaining FGFR expression at the levels required for the regulation of neuronal migration and positioning within the cortex [93]. Studies in humans have found six *de novo CDH2* missense mutations in EC5 associated with a variety of neurodevelopmental disorders, ranging from subtle disruptions of craniofacial morphogenesis and attention deficit to severe autistic disorders and intellectual disabilities [94]. Some of these mutations in *CDH2* are in the central region of EC5 [94], suggesting that disruption of CDH2-mediated regulation of FGFR signaling affects neural development by perturbing cell adhesion and cell signaling pathways regulating neuronal migration and neural circuit formation in the cerebral cortex, which may explain the significantly higher negative selective pressure exerted on the EC5 of CDH2 during the evolution of primates.

Phylogenetic analysis divides CDH type II into four groups; three of them coincide with their grouping based on heterophilic binding specificity, and the fourth matching ungrouped cadherins, suggesting that binding affinity has influenced their evolution [12,19,24]. While type I cadherins are important components of adherens junctions in different cell types

[69,77], the role of type II cadherins in the formation and regulation of adhesion junctions is not entirely understood, and the morphogenic consequences of silencing single genes are subtle as compared to the severe malformations observed in *Cdh1* and *Cdh2* null mice [75–77]. Nevertheless, CDH type II are primarily expressed in the CNS and contribute to various developmental processes, including rostrocaudal segmentation of the neural tube, formation of boundaries between anatomical structures, and specific neuronal sorting into distinct cell groups and layers [30–32,95,96]. *Cdh6* and *Cdh8* have mutually exclusive expression patterns between fore and midbrain, and their co-silencing disrupts tissue boundary formation. Triple knockout of Cdh6/Cdh8/Cdh11 causes severe neurodevelopmental defects, including cranial exencephaly, defective neurulation, and compartmentalization of the neural tube [32]. Because distinct neuronal groups express varying combinations of type II cadherins, they are thought to contribute to a molecular code involved in the specification of neuronal connectivity [71,97,98]. In the mouse retina, *Cdh6, Cdh7, Cdh8, Cdh9, Cdh10, Cdh18* are expressed individually and in combinations by neurons forming functional neuronal circuits that detect moving objects and color [34,99]. Perturbing CDH expression in the retina causes defective axonal targeting and impairs neural circuit formation, resulting in altered visually evoked responses [34,35]. Heterophilic binding between Cdh6, Cdh9, and Cdh10 expressed on apposing neuronal terminals within the hippocampus are required for high-magnitude long-term potentiation, a mechanism thought to underlie learning and memory [100,101]. Furthermore, both *CDH9* and *CDH10* have been found associated with autism spectrum disorders (ASD) [102], suggesting that genetic variations of these cadherins impair social behavior. Similarly, *Cdh8* knockout mice show defective sensory synaptic transmission [33], and *CDH8* microdeletions and polymorphisms in humans are also associated with ASD [103]. Group B cadherins are primarily expressed in neural tissues in both mice and humans and have been under high negative selection in primates. *CDH7* and *CDH12* have been associated with bipolar disorders and *CDH18* was found to be associated with schizophrenia [104–107]. Like ASD, these psychiatric disorders negatively affect social behaviors and reproductive fitness, which may explain the high level of negative selective pressure observed on cadherins highly expressed in neural tissues. In contrast to the cadherins mentioned above expression of *CDH24* is similarly distributed among neural and non-neural tissues. It has not been associated with any human disease and shows the highest number of non-synonymous substitutions among grouped type II cadherins. The absence of a phenotype impairing neural development and function may explain the lower negative selective pressure exerted on this gene. Similarly, the ungrouped *CDH5* and *CDH19* show the highest number of non-synonymous substitutions and dN/dS ratios among type II cadherins in primates. *CDH5* is primarily expressed in endothelial cells and has a similar rate of change as *CDH1*. Both genes have high essentiality scores due to the embryonic lethality caused by germ-line deletions in mice. However, both genes show the highest rates of change within CDH and the lowest expression levels in the CNS. Finally, the atypical *CDH13* has a different protein structure, lacking a CD and having an EC1 with a unique fold not observed in any other cadherin. *CDH13* is highly expressed in the CNS and has been associated with ASD and drug abuse [108–110]. In primates, *CDH13* shows a rate of evolution similar to the one observed in grouped type II cadherins that are highly expressed in the CNS. Therefore, the expression pattern of most type II cadherins and their low rate of change observed in primates suggests that the high negative selective pressure exerted on these genes is related to their importance in CNS development and function.

Factors that may deviate the rate of molecular evolution from neutrality include gene locus, essentiality, and expression pattern. Analysis of dN and dN/dS ratios between human and non-human primates of genes expressing transmembrane proteins located near each

cadherin gene show higher values than the ones observed in cadherins, suggesting that the low rate of change observed in cadherins is not related to particular characteristics of their chromosomal locus. Highly expressed genes tend to be under higher negative selective pressure possibly due to their relevance for organismal fitness [60–62]. This phenomenon is known as the expression level—evolutionary rate (E-R) anticorrelation [54–59]. However, genes coding N-glycosylated proteins deviate from the E-R anticorrelation, and instead show a positive correlation between the rate of evolution and expression levels, suggesting that factors related to protein processing and quality control mechanisms can diminish the negative selective pressure associated with the essentiality of the gene [45,51,52]. CDH are transmembrane glycoproteins expressed on the cell surface, and therefore, their rate of evolution is expected to positively correlate with their expression level [51,52]. Indeed, a positive correlation is observed between cadherins' evolutionary rate and their expression level in most human and mouse tissues; however, a negative correlation is observed with cadherins expression levels in neural tissues, indicating that cadherins highly expressed in the CNS have been under higher negative selective pressure, even though most proteins with a similar processing and subcellular localization show a positive correlation between the two. Therefore, the phenotypes observed in type II cadherins null and mutant animals described above, and the impairments observed in human behavior associated with type II cadherins mutations and polymorphisms may be subtle and difficult to detect but can significantly affect the assembly of neural circuits having deleterious effects on animal fitness, particularly in organisms with complex CNS and social behaviors.

The emergence of specialized forms of paracrine signaling associated with an intercellular adhesion junction has been a centerpiece in the evolution of the CNS, requiring the formation and fine-tuning of diverse modalities of interneuronal communication. Synapses are comprised of two distinct compartments with different molecular machinery, functions, and origins [111]. The presynaptic apparatus is thought to derive from the vesicular fusion mechanism for exocytosis, while the postsynaptic signaling complex may have arisen from cell surface sensing receptors activating diverse intracellular pathways [111–113]. A unique feature of the synapse is the precise apposition between the presynaptic active zone for neurotransmitter release and the postsynaptic specialization containing clustered receptors coupled to signal transduction. In the majority of the synapses found in the CNS, the pre and postsynaptic compartments are held together by the synaptic junction [114], providing the structural support needed for fast, precise, and efficient chemical neurotransmission. In addition, structural synaptic plasticity is associated with physiological adaptations of neural circuits relevant to cognitive functions. Although the evolutionary origin of the molecular mechanisms involved in synaptogenesis and synaptic plasticity have not been entirely elucidated, it is possible that cadherins were incorporated into the synapse soon after this specialized structure of communication emerged. Therefore, CDH contributing to morphogenic and physiological processes involved in the formation and function of neural circuits became required for the fitness of the organism and may have influenced the strength of negative selection observed on cadherins expressed in the CNS. The increase in diversity of cadherins coinciding with the expansion of molecular mechanisms involved in neurotransmitter release and in the detection of chemical signals may have contributed to the rise in variety, specificity, and complexity of neural circuits found in primates. This may explain the association of cadherin variants and mutations with psychiatric disorders having a negative impact on social behaviors and reproductive fitness. Therefore, the understanding of the evolution of CDH helps to elucidating the molecular underpinnings of this sophisticated form of intercellular communication and its relevance for the fitness and evolution of species.

## Supporting information

**S1 Table. Species and their time of divergence in Million Years Ago (MYr) based on A. Purvis Phyl Trans Royal Soc 348:405 (1995).**
(PDF)

**S2 Table. Protein and transcript accession numbers of amino acid and nucleotide sequences.**
(PDF)

**S3 Table. Codon-based Z-test of selection of CDH in non-human primates vs. humans.** Probability of rejecting the null hypothesis of strict neutrality (dS = dN) in favor of negative selection (dN<dS).
(PDF)

**S4 Table. Spearman's correlation analysis of CDH dN/dS values (New world monkeys to *H. sapiens*) and LOEUF score.**
(PDF)

**S1 File. File containing tables with the raw data, the data used for the plots presented in Figs 1 to 6, and the results of the statistical analysis.** The content of each Table is described in their heading.
(PDF)

## Acknowledgments

We thank the Allen Institute for making their data publicly available and for providing technical support. We thank Bobby Habig, Associate Professor at Mercy University for his critical reading and helpful comments to the manuscript.

## Author Contributions

**Conceptualization:** Juan L. Brusés.

**Data curation:** Max Petersen, Fredy Reyes-Vigil, Juan L. Brusés.

**Formal analysis:** Max Petersen, Fredy Reyes-Vigil, Marc Campo, Juan L. Brusés.

**Funding acquisition:** Juan L. Brusés.

**Investigation:** Max Petersen, Fredy Reyes-Vigil, Juan L. Brusés.

**Methodology:** Marc Campo, Juan L. Brusés.

**Project administration:** Juan L. Brusés.

**Resources:** Juan L. Brusés.

**Supervision:** Juan L. Brusés.

**Validation:** Marc Campo, Juan L. Brusés.

**Visualization:** Marc Campo, Juan L. Brusés.

**Writing – original draft:** Juan L. Brusés.

**Writing – review & editing:** Marc Campo, Juan L. Brusés.

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
