## [Decision Letter · Decision Letter 0]

3 Sep 2024

PONE-D-24-25424Classical cadherins evolutionary constraints in primates is associated with their expression in the central nervous system

PLOS ONE

Dear Dr. Brusés,

Thank you for submitting your manuscript to PLOS ONE and sorry for the delay in returning a decision.  I had trouble getting a second reviewer, so I reviewed it myself.  After careful consideration, we feel that it has merit but does not fully meet PLOS ONE’s publication criteria as it currently stands. Therefore, we invite you to submit a revised version of the manuscript that addresses the points raised during the review process.

First, please address the reviewer's comments point by point.  In my own review of the manuscript three points came up, the extend they require "major" review is unclear.  First, the abstract can be dramatically shortened to focus on the key observations; now it is too detailed.  Second (and minor), I would avoid definitive statements that go beyond observations throughout the ms., e.g. line 98 instead of "cadherins ... direct cell migration" better to say "cadherins ...  influence cell migration". This is particular important given that results from mouse mutations do not always translate easily to other (human) organisms. Consider, for example, the differences between early development in mouse and human.   

The final point is that I was surprised that the authors did not include genomic sequencing data from human, for example the ClinVar data available through gNomad (link to Cdh1  https://gnomad.broadinstitute.org/gene/ENSG00000039068?dataset=gnomad_r2_1).  How does  the diversity of pathogenic variants in various Catherine support or lead to rethinking of your conclusions?

We look forward to receiving your revised manuscript.

Kind regards,

Michael Klymkowsky, Ph.D.

Academic Editor

PLOS ONE

Reviewers' comments:

Reviewer's Responses to Questions

**Comments to the Author**

1. Is the manuscript technically sound, and do the data support the conclusions?

Reviewer #1: Yes

2. Has the statistical analysis been performed appropriately and rigorously? 

Reviewer #1: Yes

3. Have the authors made all data underlying the findings in their manuscript fully available?

Reviewer #1: Yes

4. Is the manuscript presented in an intelligible fashion and written in standard English?

Reviewer #1: Yes

5. Review Comments to the Author

Reviewer #1: This paper explores the evolution of cadherins in primates, finding that CDH2, CDH4, and most type II cadherins experience higher negative selective pressure, particularly due to their expression in the CNS. The research suggests that this strong selection is linked to their role in CNS development, which may have been crucial in primate evolution. The study also finds that while gene essentiality does not align with cadherin evolution rates, expression levels do.

To conduct this study, the authors examined evolutionary changes in classical cadherin orthologs across various primates by performing pairwise comparisons of synonymous and non-synonymous nucleotide substitutions to assess selective pressures on different cadherins. The ratio of non-synonymous to synonymous substitutions (dN/dS) was analyzed to measure the strength of selection. Additionally, the study evaluated gene essentiality and conducted Spearman's correlation analysis to explore the relationship between cadherin expression levels and evolutionary rates, with a particular focus on the central nervous system (CNS).

This work utilizes publicly available data on gene and protein sequences, expression levels of various classical cadherins, and gene essentiality information based on the existence of mutant mouse models for these genes to conduct the analyses mentioned. I find the work very interesting and relevant to the field. Although the paper is well-written, since it is exclusively a data analysis paper, the reading can be somewhat dense due to the extensive analysis of numerous genes and species. However, some of the figure legends are quite brief considering the amount of information each graph contains. Apart from these general considerations, I have only a few minor concerns that I believe the authors could easily address:

• On page 11, line 231, the authors discuss that “EC5 has been under higher negative selection in CDH2 and CDH4, possibly due to a functional role of this domain in these cadherins.” However, they do not provide supporting data for this statement regarding the function of this EC in those cadherins. The authors should elaborate on this issue according to the literature.

• On page 32, line 657, they discuss the association of CDH9 and CDH10 with ASD and, in line 660, how Cdh8 microdeletions and polymorphisms in humans are associated with the same syndrome. In both cases, they cite two papers (references 101 and 102). However, reference 101 relates only to CDH8, and 102 to CDH9 and CDH10.

• Authors should clarify the election of the different statistical test for the different analyses.

6. PLOS authors have the option to publish the peer review history of their article (what does this mean?). If published, this will include your full peer review and any attached files.

Reviewer #1: No

---

## [Author Response · Author response to Decision Letter 0]

18 Oct 2024

Line numbers mentioned below correspond to the final Unmarked version of the manuscript.

Reviewer 1

1.Comment: "First, the abstract can be dramatically shortened."

Response: As requested, the Abstract has been edited and shortened from 422 to 339 words. 

2.Comment: "Second (and minor), I would avoid definitive statements."

Response: Adjectives suggesting the involvement of the molecule in a particular process rather than a direct participation have been used throughout the manuscript. 

3. Comment: "The final point is that I was surprised that the authors did not include genomic sequencing data from human, for example the ClinVar data available through gNomad."

Response: Data from the gnmad database has been processed and added to the manuscript. In particular, the LOEUF scores of cadherins were correlated with the dN/dS ratios and with the level of CDH expression in 387 brain mouse cells. In both cases, the results agree with our previous results in that LOEUF scores positively correlate with dN/dS values, and negatively correlate with scRNA expression levels. The results of these analyses are presented in S4 Table, Fig 6 pane D, and S5 File. The raw data used for the plot is included in the S5 File. A description of these results starts in line 519.

Unfortunately, the number of clinical variants available in ClinVar varies substantially among CDH from over 500 in CDH1 to none in various type II CDH, making it difficult to conduct comparisons among CDS. 

Finaly, in addition to the Supplementary Information presented in Tables S1 to S4 mentioned in the text, tables containing all raw data, data plotted in the figures, and results of the statistical analyses are compiled and included in the S5 File.

Reviewer 2:

1.Comment: "On page 11, line 231, the authors discuss that “EC5 has been under higher negative selection in CDH2 and CDH4, possibly due to a functional role of this domain in these cadherins.” However, they do not provide supporting data for this statement regarding the function of this EC in those cadherins. The authors should elaborate on this issue according to the literature.

Response: A paragraph addressing the possible role of EC5 in CDH2 function has been added to the Discussion starting in line 608.

2.Comment: "On page 32, line 657, they discuss the association of CDH9 and CDH10 with ASD and, in line 660, how Cdh8 microdeletions and polymorphisms in humans are associated with the same syndrome. In both cases, they cite two papers (references 101 and 102). However, reference 101 relates only to CDH8, and 102 to CDH9 and CDH10."

Response: The error has been corrected starting in line 656. Wang et al 2009 is used as a reference for the association CDH9 and CDH10 with ASD, and Pagnamenta et al 2011 for the association of CDH8 microdeletions with ASD.

3. Comment: "Authors should clarify the election of the different statistical test for the different analyses."

Response: A description of the selection of the statistical tests is described in the Methods section line 169.

---

## [Decision Letter · Decision Letter 1]

24 Oct 2024

Classical cadherins evolutionary constraints in primates is associated with their expression in the central nervous system

PONE-D-24-25424R1

Dear Dr. Brusés,

We’re pleased to inform you that your manuscript has been judged scientifically suitable for publication and will be formally accepted for publication once it meets all outstanding technical requirements.

Kind regards,

Michael Klymkowsky, Ph.D.

Academic Editor

PLOS ONE

Additional Editor Comments (optional):

Reviewers' comments:

Reviewer's Responses to Questions

**Comments to the Author**

1. If the authors have adequately addressed your comments raised in a previous round of review and you feel that this manuscript is now acceptable for publication, you may indicate that here to bypass the “Comments to the Author” section, enter your conflict of interest statement in the “Confidential to Editor” section, and submit your "Accept" recommendation.

Reviewer #1: All comments have been addressed

2. Is the manuscript technically sound, and do the data support the conclusions?

Reviewer #1: Yes

3. Has the statistical analysis been performed appropriately and rigorously? 

Reviewer #1: Yes

4. Have the authors made all data underlying the findings in their manuscript fully available?

Reviewer #1: Yes

5. Is the manuscript presented in an intelligible fashion and written in standard English?

Reviewer #1: Yes

6. Review Comments to the Author

Reviewer #1: The authors have properly addressed my previous concerns about the manuscript, including the necessary information and correcting a previously detected error.

7. PLOS authors have the option to publish the peer review history of their article (what does this mean?). If published, this will include your full peer review and any attached files.

Reviewer #1: No

---

## [Editor Report · Acceptance letter]

13 Nov 2024

PONE-D-24-25424R1 

PLOS ONE

Dear Dr. Brusés, 

I'm pleased to inform you that your manuscript has been deemed suitable for publication in PLOS ONE. Congratulations! Your manuscript is now being handed over to our production team.

Kind regards, 

on behalf of

Dr. Michael Klymkowsky 

Academic Editor

PLOS ONE